# Extensive Limb Lengthening for Achondroplasia and Hypochondroplasia

**DOI:** 10.3390/children8070540

**Published:** 2021-06-24

**Authors:** Dror Paley

**Affiliations:** Paley Orthopedic & Spine Institute, 901 45th St., West Palm Beach, FL 33407, USA; dpaley@paleyinstitute.org

**Keywords:** achondroplasia, hypochondroplasia, dwarfism, short-limb, short stature, FGFR3, skeletal dysplasia, genetic condition, extensive limb lengthening

## Abstract

Extensive limb lengthening (ELL) was completed in 75 patients: 66 achondroplasia and 9 hypochondroplasia. The average lengthening was 27 cm for achondroplasia (12–40 cm) and 17 cm for hypochondroplasia (range 10–25 cm). There were 48 females and 27 males. Lengthening was done either by 2-segment (14 patients; both tibias and/or both femurs) or by 4-segment lengthenings (64 patients; both femurs and tibias at the same time). Most patients also had bilateral humeral lengthening. Patients had 2 or 3 lower limb lengthenings and one humeral lengthening. Lengthenings were either juvenile-onset (31), adolescent-onset (38) or adult-onset (6). The average age at final follow-up was 26 years old (range 17–43 years). There were few permanent sequelae of complications. The most serious was one paraparesis. All patients returned to activities of normal living and only one was made worse by the surgery (paraparesis). This is the first study to show that ELL can lead to an increase of height into the normal height range. Previous studies showed mean increases of height of up to 20 cm, while this study consistently showed an average increase of 30 cm (range 15–40 cm) for juvenile-onset and mean increase of 26 cm (range 15–30 cm) for adolescent-onset. This results in low normal height at skeletal maturity for males and females. The adult-onset had a mean increase of 16.8 (range 12–22 cm). This long-term follow-up study shows that ELL can be done safely even with large lengthenings and that 4-segment lengthening may offer advantages over 2-segment lengthening. While all but the more recent cases were performed using external fixation, implantable limb lengthening promises to be an excellent alternative and perhaps an improvement.

## 1. Introduction

Limb lengthening for achondroplasia is controversial [1,2,3]. Modern techniques of limb lengthening, using distraction osteogenesis, have been able to add significant length to the lower and upper limbs of children and adults with achondroplasia and hypochondroplasia [1,2,3,4,5,6]. The long-term results of these treatments have remained unknown. There has been concern that while the increased stature may be obtained, the patients will eventually develop degeneration of their joints or other long-term complications [2,3]. The purpose of this study is to determine what are the long-term results of extensive limb lengthening (ELL) in achondroplasia and hypochondroplasia.

From 1987 to 1997, all the cases of ELL performed by the author were by two-segment lengthenings; bilateral tibial lengthening [4] (double-level lengthening with external fixation in most cases) followed by single-level bilateral femoral lengthening (Figure 1).

The usual goal in the tibias was 15–16 cm and the goal in the femurs was 10–12 cm. Since 1997, the author switched to simultaneous 4-segment lengthening of both femurs and both tibias at the same time using external fixation [5] (Figure 2, Figure 3, Figure 4, Figure 5, Figure 6 and Figure 7). The femurs were lengthened to a maximum goal of 8 cm and the tibias to 7 cm, for a total gain in height of 15 cm. This method was later modified to do bilateral femoral lengthening with implantable lengthening nails simultaneous with bilateral tibial lengthening with external fixation (Figure 8 left). This was referred to as 4-segment hybrid lengthening (Figure 8 middle). Since 2014, with the advent of shorter and smaller diameter implantable lengthening nails, 4-segment femur and tibia all implantable nail lengthenings were performed (Figure 8, right) [6,7].

The age of starting the first lengthening was 12 yrs. in girls and 13 yrs. in boys. This was referred to as the adolescent-onset (age of first lengthening 12–17 yrs.). In 1997, we also began to lengthen at age 7 yrs. for girls and 8 yrs. for boys and referred to this as the juvenile-onset (age of first lengthening 7–11 yrs.). Patients who presented for the first lengthening at age 18 or over were referred to as adult-onset lengthening. 

The cumulative goal of serial lengthenings was to bring the child to low normal height for their sex at skeletal maturity. This is 4′11″ in girls and 5′4″ in boys (2.5th percentile). To achieve this in achondroplasia for both boys and girls usually required between 30 and 40 cm total height gain by skeletal maturity. To achieve this in hypochondroplasia usually required between 15 and 25 cm total height gain. With two-segment lengthenings, total height gain was usually 25 cm with one bilateral tibial (15 cm) followed by one bilateral femoral lengthening (10 cm). With 4-segment lengthening, the lengthening goal at the adolescent age was 8 cm from the femurs and 7 cm from the tibias (total 15 cm) with each adolescent 4-segment lengthening. Therefore two 4-segment lengthenings usually achieved 30 cm height gain. If the 4-segment lengthening began at the juvenile age group, the lengthening goal was 5–6 cm lengthening in the femurs and 4–5 cm lengthening in the tibias for a total height gain of 10 cm. Adding this 10 cm to the 30 cm achievable in adolescence results in a total of 40 cm total height gain with three lower limb lengthenings with the juvenile-onset strategy. 

The theoretical advantage of smaller 4-segment lengthenings vs. larger 2-segment lengthening is multifold. 4-segment limits the total lengthening per bone segment to no more than 8 cm, and 2-segment lengthening lengthens segments up to 15 cm in the tibia through double-level bone cuts and 10 cm through a single bone cut. The 4-segment technique is like a double-level lengthening spread across two separate bone segments. This reduces pain, risk of nerve injury and muscle contractures [8]. In the juvenile age group, there is also the consideration of growth inhibition following lengthening. Growth inhibition has been shown to be related to the amount of lengthening, especially in the tibia [9]. Therefore, restricting total lengthening in the tibia to 5 cm and lengthening the femur 5 cm achieves a double-level lengthening of femur and tibia of 10 cm. It is felt that this will reduce the risk of growth inhibition at a young age. 

## 2. Materials and Methods

This study was a retrospective x-ray and records review of all achondroplastic and hypochondroplastic patients treated by ELL by the author since 1987. Of those, only the ones whose follow-up is known until skeletal maturity and beyond and who completed all the serial lengthenings planned were included in this study. There were 106 patients treated between 1987 and 2020, a 33-year period. Of these patients, 75 completed all lengthenings planned and were followed into adulthood. Thirteen were lost to follow-up after completing all planned lengthenings and 18 have not completed all planned lengthenings and are currently under treatment or follow-up between lengthenings. This study will be of the 75 that completed all stages of lengthening according to juvenile-, adolescent- or adult-onset strategies. 

There were 48 females and 27 males. The clinical diagnosis was achondroplasia in 66 and hypochondroplasia in 9. The total amount of lengthening performed was an average of 27 cm (range 12–40 cm) for achondroplasia and an average of 17 cm (range 10–25 cm) for hypochondroplasia. 

## 3. Results

Lengthening was performed as either 2-segment, bilateral femoral or bilateral tibial; or 4-segment, simultaneous bilateral femoral and tibial. 2-segment tibial followed by 2-segment femoral lengthening was done in 14 patients. There were 64 patients who had 4-segment lengthenings. Three patients had both 4-segment and 2-segment lengthenings at different times. Of the 4-segment lengthenings, 14 had three 4-segment lengthenings, 38 had two 4-segment lengthenings, and 12 had one 4-segment lengthening (Table 1). 

2-segment lengthenings were done either with external fixators (exfix) 9 times, or with implantable lengthening nails (infix) four times. 4-segment lengthenings were done either with external fixators for both femurs and both tibias (all exfix) 107 times, external fixators on both tibias and implantable lengthening nails for both femurs (hybrid exfix-infix) five times, or both femurs and both tibias with implantable lengthening nails (all infix) 16 times. Lengthening of the humerus [10,11] for between 10 and 12 cm was done in all patients with achondroplasia and all but two patients with hypochondroplasia for between 10 and 12 cm (exfix in all but two cases which were done with infix) (Table 2). 

The average age at final follow-up was 26 years old (range 17–43 years). This represented a follow-up average of 13 years in the adult-onset group, vs. 10 years in the combined adolescent- and juvenile-onset groups (range 2–33 years) (Table 3).

While there were various complications during lengthening [4,12], for the purpose of this end-result study, we only tabulated the ones that led to permanent sequalae. One female patient, aged 16, developed paraparesis upon waking up from anesthesia at the time of removal of the external fixators after a successful second 2-segment lower-limb lengthening (total length gain was 25 cm). She did not have any neurologic dysfunction during the distraction or consolidation phases. She did have a recognized thoracolumbar kyphosis which her neurosurgeon elected not to treat since she was asymptomatic. The paraparesis was treated by decompression and fusion and did improve, although she was left with some permanent lower-limb weakness. A second patient developed RSD after humeral lengthening. This eventually resolved after decompression of ulnar and radial nerves.

Other long-term findings not directly related to lengthening included one patient who committed suicide several years after finishing lengthening. He had always expressed satisfaction with the lengthening. He apparently struggled with depression while studying in university, unbeknownst to his parents and friends. Three patients had significant hip dysplasia. Two of these were treated by periacetabular osteotomy immediately after completing lengthenings. A third patient went on to develop arthritis of one hip which will likely require a total hip replacement. No other patients developed degenerative joints of hips, knees, or ankles. There was no evidence of radiographic deterioration of the joints after recovering from lengthening. Five patients developed spinal stenosis requiring surgery, either prior to, during treatment, or at a later date remote from the lengthening.

Patients started lengthening between ages 6–11 (juvenile-onset 31), 12–17 years (adolescent-onset 38), or 18 years and older (adult-onset 6) (Table 3 and Table 4). 

All patients had completed high school, and many went on to higher education. All patients had returned to normal activities of daily living. Many were also participating in sporting activities. No patient was negatively impacted by the lengthening physically or psychologically, except the one patient who became paraparetic. The parents of the one patient who committed suicide were emphatic that his depression was not related to his previous lengthening surgery, which they felt was a very positive event in his life. Many patients were employed, and some were married and had children. Every patient said they would do this again despite the hardships. This included the patients with adult-onset ELL surgery.

## 4. Discussions

There have been numerous reviews of lengthening results in achondroplasia and hypochondroplasia [13,14,15,16,17,18,19,20]. Most focused on the short-term results and specific lengthening complications. Park et al. 2015 from Korea [13] reported on 28 achondroplastic patients who underwent lengthening as adolescents (bilateral tibias followed by bilateral femurs). The average height gain was 18.2 cm (8.4 femurs, 9.8 tibias). Fewer complications were found with lengthening of the femur than with lengthening of the tibia. Chibule et al. from India [14] presented 8 cases with an average of 15.4 cm lengthening of the femurs, 9.9 cm of the tibias and 9.6 cm of the humeri. Donaldson et al. from the UK [15] reported on 10 achondroplastic children who lengthened a total average of 20 cm with good final results. Yasui et al. from Japan [16] published on 35 patients with achondroplasia and hypochondroplasia lengthened with external fixation. The average total length gain was 14.3 cm (7.2 cm femurs, 7.1 cm tibias). Aldegheri et al. from Italy [17] reported on 100 patients with achondroplasia and hypochondroplasia who underwent an average of 20.5 cm lengthening for the former and 17.5 cm for the latter. Lee and Chow from Hong Kong [18] published on 8 patients with an average gain of 5.2 cm in the femurs and 5.8 cm gain in the tibias. Schiedel and Rodl from Germany [19] did a meta-analysis of 172 patients reported in the literature. The average length gain ranged from 5.7 to 20.5 cm. Leiva-Gea et al. from Spain [20] reported on the use of 4-segment lengthening for achondroplasia in 17 patients. The average height gain was 14.4 cm. 

The average male height in achondroplasia is 132 cm with the low end of height range being 118 cm and the high end 146 cm [21,22]. The low end of normal height range for a male is 160 cm. The difference between the low-end, average and high-end height for a male with achondroplasia vs. the low-end normal height for a male is 42 cm, 28 cm, and 14 cm respectively. 

The average female height in achondroplasia is 125 cm with the low end of height range being 114 cm and the high end 138 cm. The low end of normal height range for a female is 150 cm. The difference between the low-end, average and high-end height for a female with achondroplasia vs. the low-end normal height for a female is 36 cm, 25 cm, and 12 cm respectively. 

What, therefore, should be the height gain goal of extensive limb lengthening? Our goal was to achieve the low end of normal height for their sex. This requires between 14 and 42 cm in males and between 12 and 36 cm in females. Using Schiedel and Rodl’s [19] metanalysis as the range of height gain from lengthening from most studies (5.7–20.5 cm) suggests that few, if any, achondroplastic patients achieve low normal height for their sex with the methods used by most authors (2-segment: bilateral femoral and bilateral tibial lengthening) [13,14,15,16,17,18]. Even the one 4-segment lengthening study by Leiva-Gea et al. [20] from Spain, only achieved 14.4 cm average height gain. In our study, the average gain was 27 cm for achondroplasia and 17 cm for hypochondroplasia. The 29 patients with achondroplasia who underwent the juvenile-onset strategy had an average lower-limb length gain of 30.1 cm (median 28 cm, range 15–40 cm). Twenty-five of the 29 gained 25 cm or greater. The 33 achondroplasia patients who underwent the adolescent-onset strategy gained an average of 26 cm (median 27 cm; range 15–30 cm). Twenty-seven of the 33 gained 25 cm or greater. The 4 adult achondroplastic patients gained an average of 16.8 cm (range 12–22 cm). It is clear that both the adolescent-onset as well as the juvenile-onset strategies gained sufficient height to reach the low normal height for their sex. Almost all patients in both groups achieved height gain of 25 cm or greater. Half the patients in the juvenile-onset group achieved gains of 30 cm or greater up to 40 cm. This study is therefore the first study to demonstrate that extensive limb lengthening for achondroplasia can result in safe increase of height to normal adult levels for each sex. In contrast, previous methods reported have fallen short of that goal [13,14,15,16,17,18,19,20]. This study demonstrates that the criticism that ELL can only produce ‘taller dwarfs’ is false [23]. ELL can add sufficient height to restore achondroplastic patients to normal adult heights for each sex in the majority of cases. Two other published studies shared this goal although it is not clear if they ever achieved it. Vilarrubias et al. did large 2-segment lengthenings of both femurs and both tibias at separate times with each lengthening between 15 and 18 cm in each bone. The complications rates were very high and the recovery times from such large lengthenings were very long [24]. They also did bilateral humeral lengthening. Peretti et al. preferred to lengthen smaller amounts but do 3 and up to four lengthening surgeries to achieve a goal of 30–35 cm [25]. Both Vilarrubias et al. and Peretti et al. recognized that the goal of lengthening should be to aim for a low normal height for the sex of the patient [24,25]. 

There are no published growth charts for hypochondroplasia. In this author’s experience, the final adult height in hypochondroplasia is 10–15 cm taller than in achondroplasia. In this study, the average length gain in hypochondroplasia was 16.8 cm (range 10–25 cm). All of these patients reached normal adult height for their sex.

Although we did not do quality of life (QOL) or psychologic analysis, prior studies of QOL have shown that despite frequent complications and despite the limited height gain as previously described, the QOL improved with ELL [26]. Batibay et al. showed that despite complications with lower limb lengthening the QOL scores improved significantly if humeral lengthening was also performed [27]. They also showed that QOL scores were higher in the lengthened group vs. a control non-lengthened achondroplasia group. Matsushita et al. showed that achieving a height greater than or equal to 140 cm was associated with higher QOL scores [28]. Fernandes et al. showed that limb-lengthening surgery helped patients in Spain integrate into society better than in the USA, where such surgery was less common [23]. Lavini et al. reported one of the few psychologic studies on patients going through ELL. They started lengthening in adolescence [29]. Their patients were motivated by a combination of being able to overcome some physical shortcomings related to their short stature and short limbs and, at the same time, improving their social life. Their study claims that these goals were achieved by ELL.

The 4-segment lengthening method is currently our preferred lengthening strategy to achieve maximum total length gain in the shortest possible time without exceeding 8 cm lengthening in any single bone segment. While this strategy was originally used with 4-segment external fixation application, it can also be done with 4-segment implantable limb-lengthening devices or with hybrid 4-segment lengthening; two segments with external and two segments with internal devices. When four intramedullary nails are inserted, we stagger the insertion by doing the two tibias, followed three weeks later by the two femurs. This is to reduce the risk of fat embolism syndrome from reaming 4 bones at one time. When the nails are exchanged for a second lengthening, the bones do not have to be reamed again, and therefore, all four rods can be inserted in one surgery. Most recently, the extramedullary lengthening with Precice nail or Precice plate has allowed us to use implantable lengthening nails in the femurs and lengthening plates in the tibias (Figure 9) [30].

This avoids reaming through the proximal tibial growth plate. Since extra-medullary application does not involve reaming, two intramedullary nails and two extramedullary nails or plates can be placed at one time. When the goal is to gain less total length, a 2-segment strategy may be preferable to a 4-segment strategy. This is more applicable to ELL for hypochondroplasia, where the lengthening goal is only 16 cm on average. In contrast, when the lengthening goal is 25–40 cm as in achondroplasia, the 4-segment strategy is preferred.

This study confirms that ELL planning should consider the total height gains desired based on prediction of adult height using the achondroplasia or hypochondroplasia growth charts [21,22,24]. To achieve the desired gain in height, the adolescent-onset strategy should be chosen if the goal is under 30 cm. The juvenile-onset strategy should be chosen if the goal is 30–40 cm. The juvenile-onset strategy also has the advantage of not allowing the achondroplastic or hypochondroplastic child to fall too far behind their peers in height. Lengthening during adolescence offers children with achondroplasia and hypochondroplasia a ‘growth spurt’ in parallel with their peers. Lengthening younger than age 7 in girls and 8 in boys risks growth inhibition and also is limited in total amount of length gain [9]. In this authors opinion, it should be avoided. Parents who want ELL should also avoid having osteotomy for genu varum at a young age since the genu varum can be corrected at the time of the juvenile-onset lengthening [31]. This is another advantage of the juvenile-onset strategy. 

With the introduction of pharmacotherapeutics for achondroplasia [32], ELL is likely to become an abandoned treatment. Since the height gain from these new drugs may not achieve full normal adult height (especially depending on what age these drugs are started, e.g., girls achieve half their lower- and upper-limb growth by age 3 yrs. and boys by age 4 yrs.; by age 1, both boys and girls have one third of their total adult limb lengths). Therefore, if the therapeutics do not start until age 4 they are unlikely to regain the inhibited growth that already occurred. For this reason, it is likely that stature lengthening for children with achondroplasia will still be needed, but to a lesser extent, and only at skeletal maturity. This may involve one 2- or 4-segment lengthening with fully implantable lengthening devices.

## Figures and Tables

**Figure 1 children-08-00540-f001:**
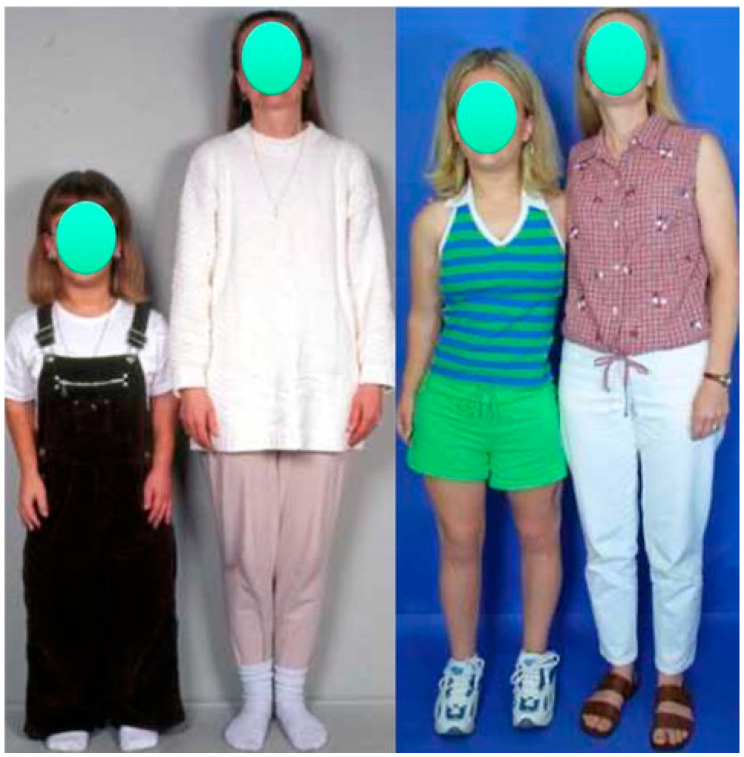
Photographs of a female patient with achondroplasia standing beside her mother at age 13 (**left**) and at age 17 (**right**). Her height increased from 4′ (122 cm) before lengthening to 5′1.5″ (156 cm) after two lower-limb 2-segment lower-limb adolescent-onset lengthenings totaling 10.25″ (26 cm). She also had 10 cm of lengthening of her humeri.

**Figure 2 children-08-00540-f002:**
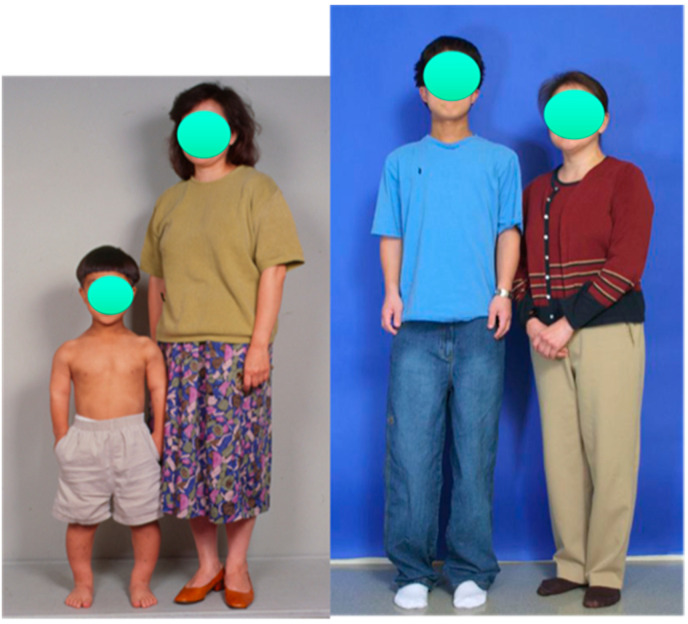
Photograph of male patient with achondroplasia standing beside his mother at age 10 (**left**) and age 16 (**right**). His height increased from 3′9″ (114 cm) before lengthening to 5′5.5″ (165 cm) after three 4-segment lower-limb juvenile-onset lengthenings totaling 15.75″ (40 cm). He also had 12 cm lengthening of his humeri.

**Figure 3 children-08-00540-f003:**
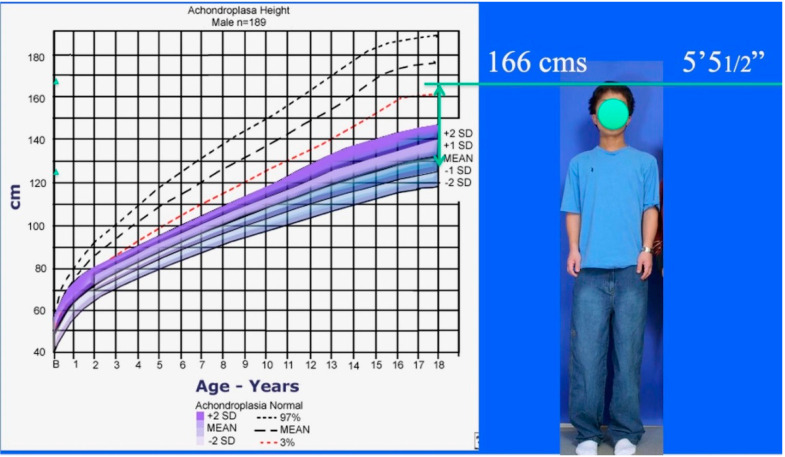
Achondroplastic male height growth chart (lower shaded) compared to normal male height growth chart (upper dashed). Achrondroplastic boy from Figure 1 standing to right of growth chart showing his 15.75″ = 40 cm lengthening added to his preoperative predicted height at maturity (120 cm). Final adult height after three 4-segment lower limb lengthenings is 5′5.5″ = 166.3 cm which is in the normal height range for an adult male.

**Figure 4 children-08-00540-f004:**
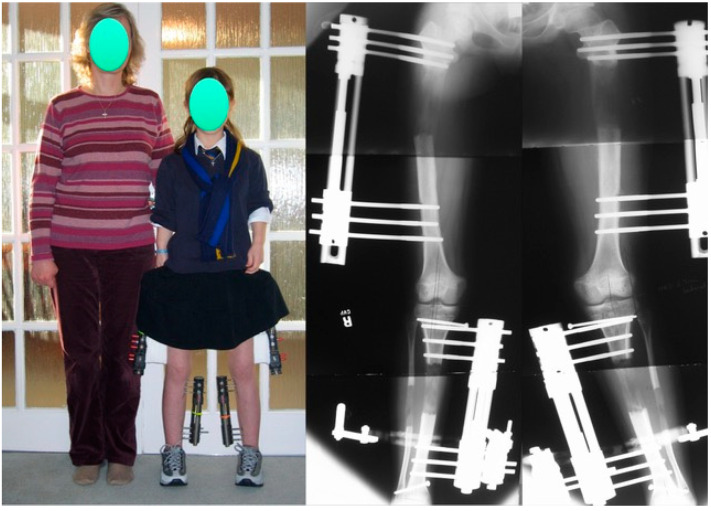
Achondroplastic girl standing next to her mother during the second of three juvenile-onset lower-limb 4-segment lengthenings. The 4 monolateral fixators are clearly visible. The radiograph to the right shows an 8 cm gain in the femurs and a 7 cm gain in the tibia-fibula.

**Figure 5 children-08-00540-f005:**
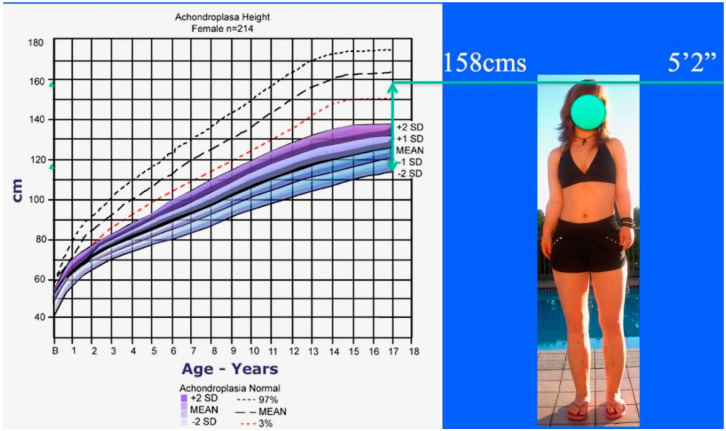
Achondroplastic female height growth chart (lower shaded) compared to normal female height growth chart (upper dashed). Achondroplastic girl from Figure 3 standing to the right of the growth chart showing her 15.75″ = 40 cm lengthening added to her preoperative predicted height at maturity (115 cm). Final adult height after three 4-segment leg lengthenings is 5′2″ = 157.5 cm, which is in the normal height range for an adult female.

**Figure 6 children-08-00540-f006:**
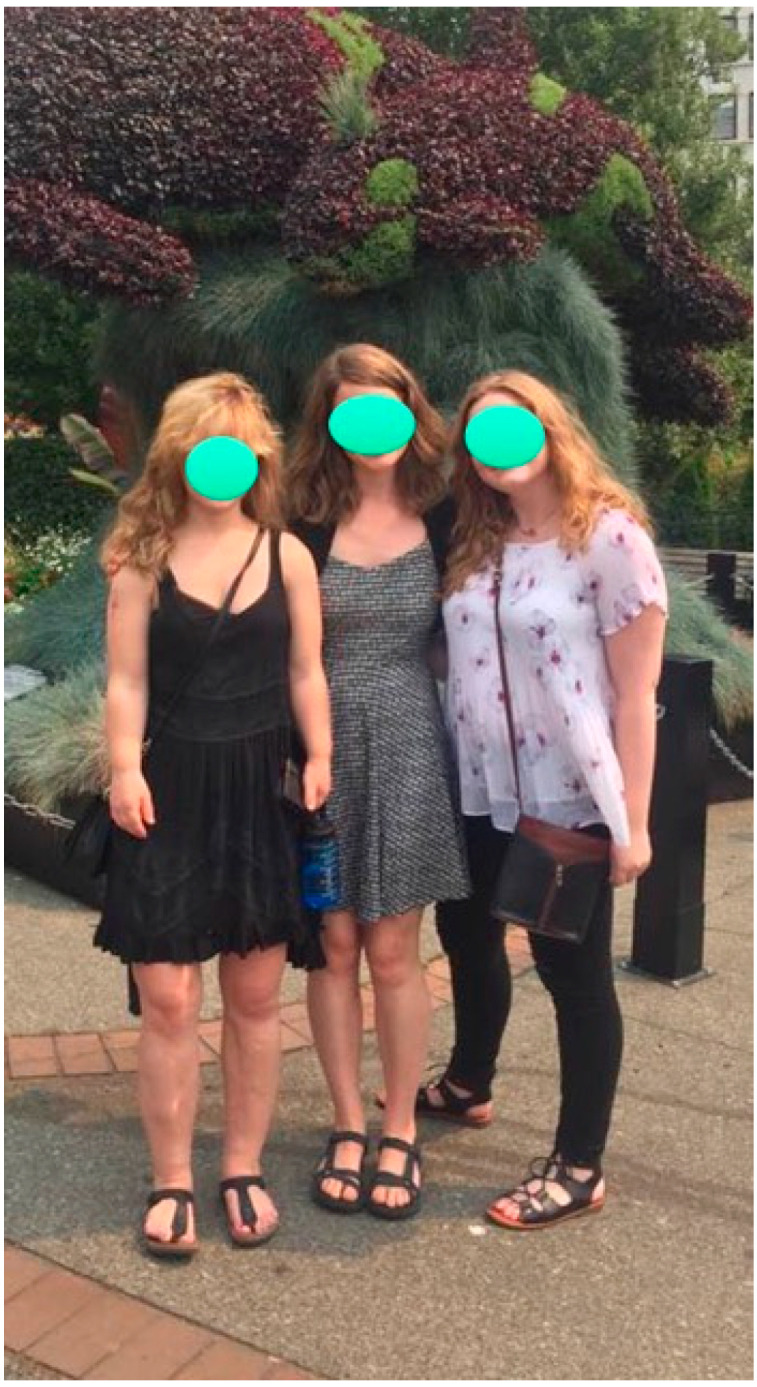
Same patient as in Figure 4 and Figure 5 standing to the right of both of her friends (left side of picture), all of whom are of normal height.

**Figure 7 children-08-00540-f007:**
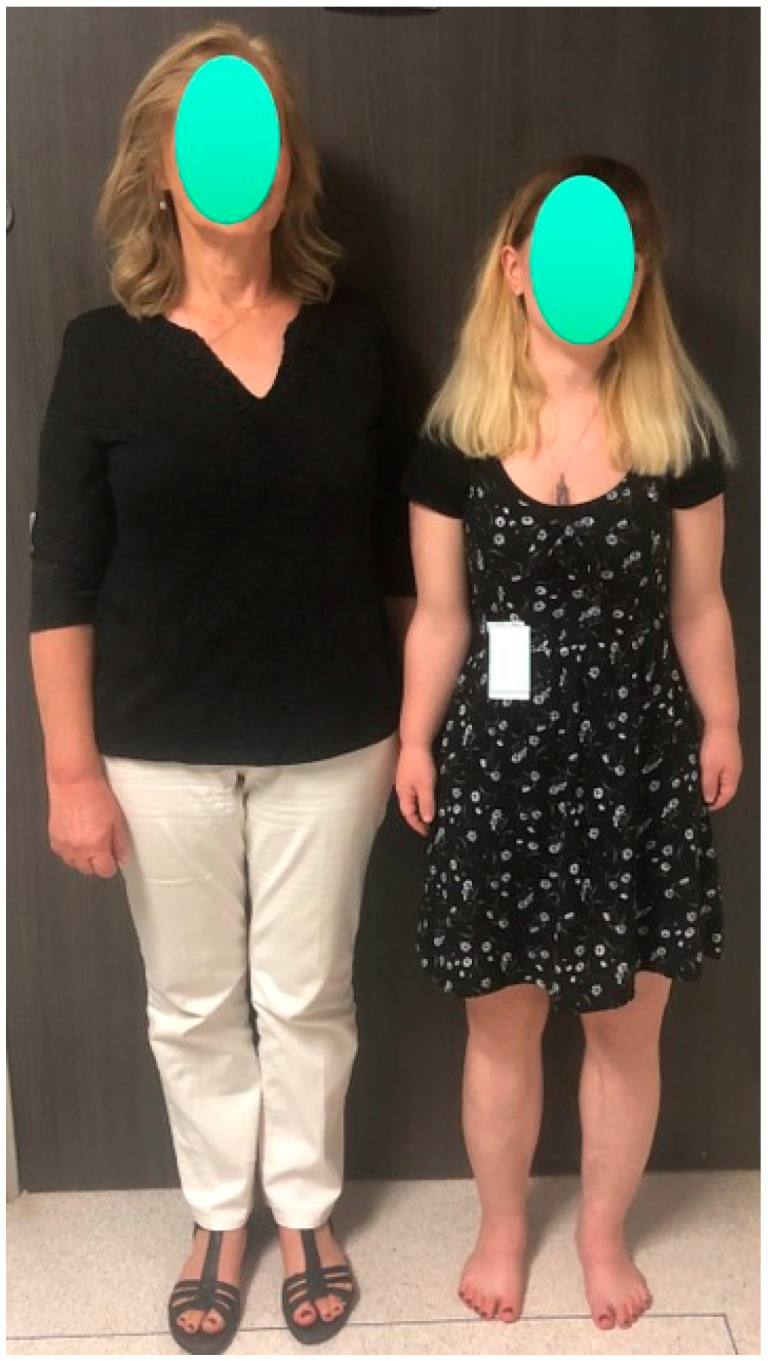
Same patient as in Figure 4, Figure 5 and Figure 6 standing next to her mother at the age of 28. She began the juvenile-onset ELL at age 7 years. The patient is 5′2″ compared to her mother who is 5′9″.

**Figure 8 children-08-00540-f008:**
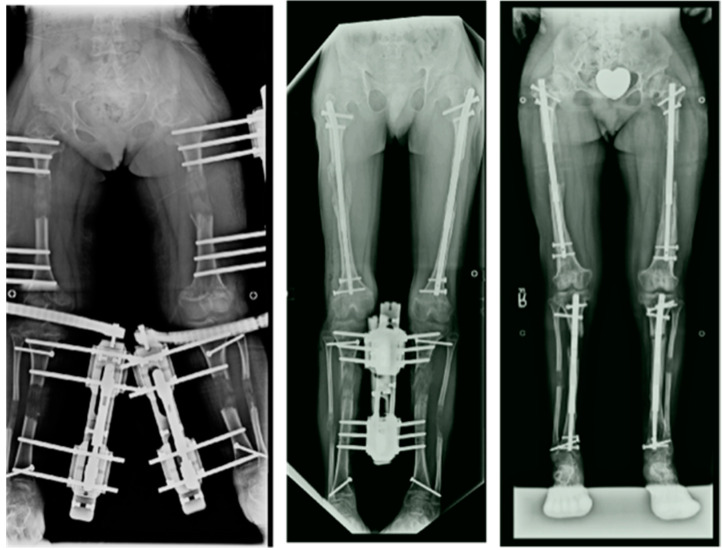
Radiographs showing 4-segment lengthening with: 4 external fixators (**left**); Hybrid, 2 implantable lengthening nails in both femurs and two external fixators (**middle**); and 4 implantable lengthening nails in both femurs and tibias.

**Figure 9 children-08-00540-f009:**
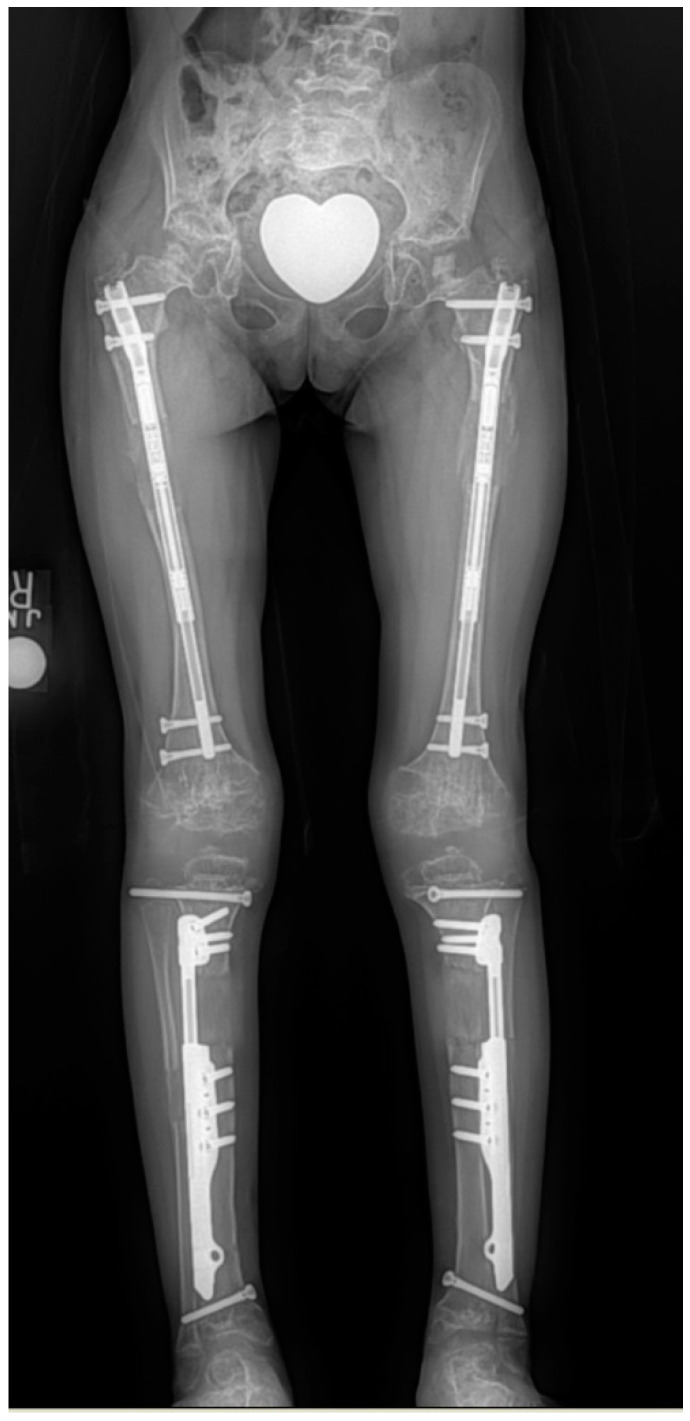
Radiograph showing InFix lengthening in a 12-years-old child with pseudo-achondroplasia using two implantable lengthening nails in the femurs and two implantable lengthening plates in the tibias.

**Table 1 children-08-00540-t001:** Lengthening types by segments (S: Segment).

Type	2-S × 2	2-S + 4-S	4-S × 1	4-S × 2	4-S × 3
Patients	14	3	14	38	12

**Table 2 children-08-00540-t002:** Lengthening types by devices.

Device	EF	Hybrid	ILN
2-S	9		4
4-S	107	5	16
Humerus	71		2

EF—External Fixator, Hybrid—EF tibias +ILN femurs, ILN—Intramedullary Lengthening Nail.

**Table 3 children-08-00540-t003:** Onset age demographics for ELL (extensive limb lengthening).

ELL Onset Age	Juvenile (7–11 yrs.)	Adolescent (12–17 yrs.)	Adult (18 yrs. & Older)
Patient #	31	38	6
M/F	13/18	12/26	2/4
Follow-up Age (yrs.)	24	27	33
Follow-up Age Range (range)	17–35	17–46	23–43

**Table 4 children-08-00540-t004:** ELL age and lengthening goal algorithm for Achondroplasia and Hypochondroplasia.

Juvenile/Adolescent	Achondroplasia	Method	Hyopchondroplasia	Method
ELL Goal	30–40 cm		15–25 cm	
Age 7–11 yrs	F 5 cm + T 5 cm	4-S	±F-5 cm + T-5 cm	4-S
Age12–13 yrs	F 5–8 cm + T 5–8 cm	4-S	F 8 cm ± T 8 cm	2-S or 4-S
Age 14–15 yrs	H 10–12 cm	2-S	H 10 cm	2-S
Age 15–16 yrs	F 5–8 cm + T 5–8 cm	4-S	T 8 cm if previous F 2-S	2-S if 2-S prior

F—Femur, T—Tibia, S—Segment.

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
