# Peer review of "Extensive Limb Lengthening for Achondroplasia and Hypochondroplasia"

_children, 2021, doi:10.3390/children8070540_

Round 1
Reviewer 1 Report
This is a generally well written MS, which reviews the author’s experience in extended limb lengthening in achondroplasia and hypochondroplasia. It apparently is submitted for a special issue edited by one of his colleagues. The author is an internationally recognized practitioner of and proselytizer for extended limb lengthening. Its use in asymmetric deformity is uncontroversial. Not so for symmetric shortening secondary to intrinsic bone dysplasias in which there is ongoing controversy about its ultimate utility.
While this review of his experience is appropriately oriented toward surgical outcomes, nonetheless I think it is incumbent on the author to more fully, if briefly, discuss the psychological aspects of this undertaking:
- What method of consent was obtained/obtainable from 7- and 8-year-olds who were members of the juvenile group?
- How was it ascertained that “no patient was negatively impacted…psychologically”?
- What antecedent assessments of psychological appropriateness was there leading up to initiation of surgery?
- Citations for this being “controversial” are only to those who, in a general sense, are advocates of ELL for dwarfing conditions. Are there no available sources addressing this controversy from alternative perspectives?
Other minor issues:
- In the methods, numbers seem to simply not add up – 75 + 13 + 18 does not equal 104. This needs to be corrected or explained. [‘There was a total of 104 patients treated between 1987 to 2020, a 33-year period. Of these patients 75 completed all lengthenings planned and were followed into adulthood. Thirteen were lost to follow-up after completing all planned lengthenings and 18 have not completed all planned lengthenings and are currently under treatment or follow-up between lengthenings.’]
- The discussion of “tall dwarfs” is a misrepresentation. My understanding is that those who have used that phrase refer not to the height of the individual but to the reality that other medical complications are still going to be prevalent within this population. Probably this should simply be deleted.
- A close parenthesis is needed on p 10 line 281.
- There are some inconsistencies in the reference format which I assume the journal will take care of.
Author Response
This is a generally well written MS, which reviews the author’s experience in extended limb lengthening in achondroplasia and hypochondroplasia. It apparently is submitted for a special issue edited by one of his colleagues. The author is an internationally recognized practitioner of and proselytizer for extended limb lengthening. Its use in asymmetric deformity is uncontroversial. Not so for symmetric shortening secondary to intrinsic bone dysplasias in which there is ongoing controversy about its ultimate utility.
While this review of his experience is appropriately oriented toward surgical outcomes, nonetheless I think it is incumbent on the author to more fully, if briefly, discuss the psychological aspects of this undertaking:QoL and psychologic references and paragraph added in.
- What method of consent was obtained/obtainable from 7- and 8-year-olds who were members of the juvenile group? consent is obtained by the parents or legal guardian. Children cannot legally give informed consent until age 18 in the US. Parents are the ones who bring their children in for a consultation. I advise them of the risks and benefits on multiple occasions prior to surgery. They make the decision for the juvenile group. Adolescents are more active in the decision making process. Although parents advise them no adolescent is ever forced to do this surgery. Most of them choose to do it. Interestingly, I don't believe that children of any age make their decisions for the correct reasons. e.g. adolescents do things to fit in and not be different. By the time they are more mature at age 25-30 years they look at life differently and make decisions for different reasons and not because of peer pressure. For these reasons the best people to make the decisions are the parents since they are old enough to make decisions as an adult and not as a child. Kids make decisions differently at every stage of childhood. This is a great topic but beyond the scope of this study. We did not employ a psychologist to evaluate our patients before surgery or to treat them after surgery.
- How was it ascertained that “no patient was negatively impacted…psychologically”? This is a very subjective statement based on my impression from talking to patients and parents at different stages. One really gets to know these families because of the lengthy nature of treatment and followup. I did not observe or document report by parents of any behavioral or psychologic issues . They appeared well adjusted. I think I wrote letters of recommendation to colleges for many patients telling the admission officer that there was nothing in college that would challenge these kids as much as I did with the surgeries I performed and treatment they underwent.
- What antecedent assessments of psychological appropriateness was there leading up to initiation of surgery?None. Its not the standard of care to need to do so.
- Citations for this being “controversial” are only to those who, in a general sense, are advocates of ELL for dwarfing conditions. Are there no available sources addressing this controversy from alternative perspectives?In the US due to the LPA (Little Peoples Association) who advocate that little people are normal and should not change, any change including ELL and even the new drugs is met with resistance. There is a history of trying lengthening with the old Wagner methods in the 1970's and early 1980' that led to so many complications that lengthening fell into ill repute. The LPA has felt it has to defend its members from being preyed upon by orthopedic surgeons. That is why such a report is so important. Reference 29, Fernandez etal discusses this specifically re the LPA vs organizations such as ALPE in Europe which support ELL. I have added this in to the discussion.
Other minor issues:
- In the methods, numbers seem to simply not add up – 75 + 13 + 18 does not equal 104. Thanks for catching this. It is 106 and this was corrected. This needs to be corrected or explained. [‘There was a total of 104 patients treated between 1987 to 2020, a 33-year period. Of these patients 75 completed all lengthenings planned and were followed into adulthood. Thirteen were lost to follow-up after completing all planned lengthenings and 18 have not completed all planned lengthenings and are currently under treatment or follow-up between lengthenings.’]
- The discussion of “tall dwarfs” is a misrepresentation. My understanding is that those who have used that phrase refer not to the height of the individual but to the reality that other medical complications are still going to be prevalent within this population. Probably this should simply be deleted.The comment is pertinent and was therefore changed to 'taller' not 'tall'
- A close parenthesis is needed on p 10 line 281. done
- There are some inconsistencies in the reference format which I assume the journal will take care of. I will allow the journal to point these out and correct or revert.
Thank you for your supportive and thoughtful review, time and effort.
Reviewer 2 Report
The review describes a serie of 75 patients who underwent lengthening procedure by external fixation or intramedullary nail. This review is of valuable information for surgeons who perform lenghtening procedure in patients not only in achondroplasia or hypochondroplasia.
The author describes the development of 4 segment lengthening from both femurs and tibias with external fixation to 4-segment lengthening with implantable nails.
It would be helpful for the readership if results were presented in a table. In addition, the author stated in the discussion his personal treatment regime of adolescent-onset and juvenile-onset lengthening procedure. I strongly suggest to provide an overview of this algorithm in form of a chart or table which will support his treatment protocol and makes it easier to read and to understand.
Author Response
The review describes a serie of 75 patients who underwent lengthening procedure by external fixation or intramedullary nail. This review is of valuable information for surgeons who perform lenghtening procedure in patients not only in achondroplasia or hypochondroplasia.
The author describes the development of 4 segment lengthening from both femurs and tibias with external fixation to 4-segment lengthening with implantable nails.
It would be helpful for the readership if results were presented in a table. Thank you for this suggestions. Tables 1,2,3 were added to make quick review of the data easier. In addition, the author stated in the discussion his personal treatment regime of adolescent-onset and juvenile-onset lengthening procedure. I strongly suggest to provide an overview of this algorithm in form of a chart or table which will support his treatment protocol and makes it easier to read and to understand. Good suggestion. I have added Table 4 which is the age and diagnosis ELL treatment alogrithm summary.